# Contamination Status of Pet Cats in Thailand with Organohalogen Compounds (OHCs) and Their Hydroxylated and Methoxylated Derivatives and Estimation of Sources of Exposure to These Contaminants

**DOI:** 10.3390/ani12243520

**Published:** 2022-12-13

**Authors:** Makoto Shimasaki, Hazuki Mizukawa, Kohki Takaguchi, Aksorn Saengtienchai, Araya Ngamchirttakul, Disdanai Pencharee, Kraisiri Khidkhan, Yoshinori Ikenaka, Shouta M. M. Nakayama, Mayumi Ishizuka, Kei Nomiyama

**Affiliations:** 1Center for Marine Environmental Studies (CMES), Ehime University, Bunkyo-cho 2-5, Matsuyama 790-8577, Japan; 2Department of Science and Technology for Biological Resources and Environment, Graduate School of Agriculture, Ehime University, Tarumi 3-5-7, Matsuyama 790-8566, Japan; 3Center for Preventive Medical Sciences, Chiba University, 6-2-1 Kashiwanoha, Kashiwa 277-0882, Japan; 4Department of Pharmacology, Faculty of Veterinary Medicine, Kasetsart University, 50 Ngam Wong Wan Rd, Lat Yao, Chatuchak, Bangkok 10900, Thailand; 5Kasetsart University Veterinary Teaching Hospital Nong Pho, Faculty of Veterinary Medicine, Kasetsart University, No. 121, Moo. 8, Ban Luek Subdistrict, Photharam District, Bangkok 70120, Thailand; 6Kasetsart University Veterinary Teaching Hospital Hua Hin, Faculty of Veterinary Medicine, Kasetsart University, Petchkasem Road, Nong Kae Subdistrict, Hua Hin District, Prachuap Khiri Khan 77110, Thailand; 7Department of Environmental Veterinary Sciences, Faculty of Veterinary Medicine, Hokkaido University, Kita 18, Nishi 9, Kita-ku, Sapporo 060-0818, Japan; 8Veterinary Teaching Hospital, Graduate School of Veterinary Medicine, Hokkaido University, N18 W9, Sapporo 060-0818, Japan; 9One Health Research Center, Hokkaido University, Kita 18, Nishi 9, Kita-ku, Sapporo 060-0818, Japan

**Keywords:** pet cat, organohalogen compounds (OHCs), polybrominated diphenyl ethers (PBDEs), polychlorinated biphenyls (PCBs), cat food, house dust

## Abstract

**Simple Summary:**

We focused on the actual exposure to OHC in serum from pet cats kept in Thailand, and on cat food and house dust as expected sources of contamination. The Thai cat sera had significantly higher levels of PBDEs than other contaminants. Especially, decabromodiphenyl ether (BDE-209) is a major contaminant of OHCs in dry cat food and house dust, which was estimated to be a source of exposure for Thai pet cats. On the other hand, the level of contamination by PCBs was lower than in other countries. Analysis of pet foods suggested that BDE-209 in pet cat serum was attributable to the consumption of dry cat food. On the other hand, house dust also contained high concentrations of BDE-209. Thus, high levels of BDE-209 in pet cat sera can be attributed to the consumption of dry cat food and house dust.

**Abstract:**

In this study, we analyzed serum samples of pet cats from Thailand and estimated the contribution to organohalogen compounds (OHCs) exposure through cat food and house dust intake. BDE-209 was predominant in cat sera and accounted for 76% of all polybrominated diphenyl ethers (PBDEs). Decabromodiphenyl ether (BDE-209) is a major contaminant in dry cat food and house dust, which has been estimated to be a source of exposure for Thai pet cats. BDE-209 is a major contaminant of OHCs in dry cat food and house dust, which was estimated to be a source of exposure for Thai pet cats. On the other hand, the level of contamination by PCBs was lower than in other countries. Analysis of pet foods suggested that BDE-209 in pet cat serum was attributable to the consumption of dry cat food. On the other hand, house dust also contained high concentrations of BDE-209. Thus, high levels of BDE-209 in pet cat sera can be attributed to the consumption of dry cat food and house dust. These results suggest that pet cats are routinely exposed to non-negligible levels of OHCs.

## 1. Introduction

Organohalogen compounds (OHCs) such as polychlorinated biphenyls (PCBs) and polybrominated diphenyl ethers (PBDEs) were widely used in industry. However, because of their persistence, bioaccumulation, and toxicity, the use of PCBs, PBDE mixtures (penta- and octa-BDEs), and deca-BDE were prohibited under the Stockholm Convention in 2004, 2009, and 2017, respectively. However, PCBs and PBDEs are still ubiquitous in the environment, and they have been detected in various wild animals [1,2]. PCBs and PBDEs are metabolized to hydroxylated PCBs and PBDEs (OH-PCBs and OH-PBDEs) by phase I cytochrome P450 (CYP) activity. They are then conjugated by uridine diphosphate-glucuronosyltransferases (UGTs), sulfotransferase, or glutathione *S*-transferase in phase II, and they are excreted in urine and feces [3,4]. However, cats (*Felis silvestris catus*) have different CYP expression patterns than other carnivorans, and low-chlorinated OH-PCBs are known to remain in the body [5,6]. This is attributed to the pseudo-genetization of UGT1A6 and 2B31, which weaken the glucuronidation capacity and excretion capacity of cats [7,8].

Pet cats are exposed to OHCs via cat food and house dust [5,9,10]. Dietary intake is considered a major exposure route for most OHCs [11,12]. The ingestion of contaminated house dust significantly contributes to exposure to PBDEs, which are a brominated flame retardant and are easily released by household appliances and textile products [13,14]. Cat food made from seafood contains naturally produced OHCs such as OH-PBDEs and methoxylated PBDEs (MeO-PBDEs) as well as PCBs and PBDEs [5,15]. OH-PBDEs structurally resemble thyroid hormones (THs) such as thyroxine (T4) and can potentially disrupt TH homeostasis [16,17] and oxidative phosphorylation [18] to elicit neurotoxicity [19]. In some species, the demethylation of MeO-PBDEs by CYPs can result in the formation of OH-PBDEs rather than the metabolism of parent PBDEs. In our previous study using cat liver microsomes, we showed that 6-OH-BDE-47 and 2′-OH-BDE-68 were produced by the demethylation of 6-MeO-BDE-47 and 2′-MeO-BDE-68, respectively [5]. Therefore, the effects of not only anthropogenic OHCs but also naturally produced compounds on the health of pet cats must be considered.

Recently, the relationship between OHC exposure and pet cat diseases has received attention. Epidemiological studies have reported that indoor rearing and intake of wet food increase the morbidity of feline hyperthyroidism (FHT), which is a common disease in elderly (>10 years old) cats [20]. These studies indicated that OHC exposure via canned food and house dust could represent the pathogenesis of hyperthyroidism. Serum concentrations of some PCB and PBDE congeners have been shown to be higher in cats with FHT than without [21]. In addition, the association between PBDE and type 2 diabetes mellitus has also drawn attention [22]. Because OHCs may pose a health risk to pet cats, monitoring the levels of OHC in cat serum, cat food, and house dust is important.

In Southeast Asia, OHC contamination caused by the import of e-waste and rapid industrialization has become a serious problem [23]. PBDE concentrations in sediments in Southeast Asian countries were generally higher than those reported for industrialized countries, and the spatial distribution of PBDEs suggested that inland sources may impact coastal areas [24]. Huge amounts of e-waste from North America have been exported to countries including Thailand, and PBDEs may have been discharged to the environment during the waste treatment [25]. Until recently, Thailand did not have strong regulations for PBDE management. Thus, PBDEs have been found in various consumer products, including electronics and furniture [25], and may have high levels of exposure to pet cats. In addition, the pet market in Thailand has seen rapid growth over the past decade; the estimated average annual growth of the pet market in Southeast Asia between 2012 and 2020 was about 7% [26]. Therefore, it is expected that attention will be focused on the environment and health of pet cats in Southeast Asia in the future. However, only limited information is available on OHC exposure for pet cats, especially in Southeast Asia. In this study, serum samples of pet cats kept in Thailand were analyzed by gas chromatography-mass spectrometer, focusing on the actual exposure to OHCs and the expected sources of contamination of cat food and house dust to pets. From these results, we inferred the main routes of exposure to OHC in pet cats in Thailand.

## 2. Materials and Methods

### 2.1. Sampling

Serum samples were collected from pet cats (19 females and 22 males) at veterinary hospitals of Bangkok (*n* = 13), Nong Pho (*n* = 10), and Hua Hin (*n* = 17) in Thailand during 2013–2014. Sampling was performed when the pet cats visited the veterinary hospitals for routine medical examinations, with the consent of their owners. Serum samples were obtained after blood centrifugation in a Venoject II plastic tube (Terumo Corporation, Tokyo, Japan). None of the cats were diagnosed with any specific disease. At the same time as the serum collection, their owners were asked to answer a questionnaire regarding the housing (indoor or outdoor), type of food, and use of cat litter. The results are summarized in Table 1.

Commercially available dry cat food (*n* = 7) and wet cat food (*n* = 7) were purchased from a local supermarket in 2013. These cat foods are commonly used in Thailand and routinely fed to pet cats, the subject in this study. The main ingredients and countries of origin according to the package labels of each cat food are summarized in Table 2. This study was collected from three urban areas to collect house dust widely in Thailand. Dust samples were collected from general households in Bangkok (*n* = 1), Sukhothai (*n* = 5), and Samut Songkhram (*n* = 1). House dust was collected by the residents using a broom or vacuum cleaner. House dust collection information based on the questionnaire is shown in Appendix A. Please note that the house dust was collected additionally and is not paired with the pet cat serum samples. The collected serum, cat food, and dust samples were stored at −20 °C before chemical analysis.

### 2.2. Chemicals

We purchased ^13^C_12_-labeled PCBs (CB-52, -95, -101, -105, -118, -138, -153, -156, -157, -167, -170, -178, -180, -189, -194, -202, -206, -208, and -209), ^13^C_12_-labeled PBDEs (BDE-47, -99, -100, -153, -154, -183, -196, -197, -207, and -209), ^13^C_12_-labeled OH-PCBs (4-OH-CB-29, 4′-OH-CB-61, 4-OH-CB-79, 4′-OH-CB-120, 4-OH-CB-107, 4′-OH-CB-159, 4-OH-CB-146, 4′-OH-CB-172, and 4-OH-CB-187), ^13^C_12_-labeled OH-PBDEs (6-OH-BDE-47, 6′-OH-BDE-99, and 6′-OH-BDE-100), and a standard mixture of 31 PBDEs (BFR-PAR) from Wellington Laboratories Inc. (Guelph, ON, Canada). Standards of 56 PCBs were purchased from Cambridge Isotope Laboratories Inc. (Tewksbury, MA, USA). In addition, 52 OH-PCBs and 15 OH-/MeO-PBDEs were obtained from Wellington Laboratories Inc., Cambridge Isotope Laboratories Inc., Accustandard Inc. (New Heaven, CT, USA) and were partially synthesized [5]. Details on the OH-PCB and OH-/MeO-PBDE congeners have been presented in previous articles [5]. Analytical-grade dichloromethane (DCM), *n*-hexane, methanol, ethanol, methyl *t*-butyl ether (MTBE), decane, and silica gel (Wako-gel S1, DX, KOH, 22% H_2_SO_4_ and 44% H_2_SO_4_) were purchased from FUJIFILM Wako Pure Chemical Industries Corporation (Osaka, Japan). Trimethylsilyldiazomethane (TMSDM) was purchased from Tokyo Chemical Industry Inc. (Tokyo, Japan). Acetone and anhydrous sodium sulfate were purchased from Nacalai Tesque Inc. (Kyoto, Japan). Bio-Beads S-X-3 was purchased from Bio-Rad Laboratories Inc. (Hercules, CA, USA).

### 2.3. Analysis of Organohalogen Compounds

The methods for analyzing OHCs in cat serum have been described elsewhere [27]. Briefly, a serum sample (0.5 mL) spiked with ^13^C_12_-labeled internal standards was denatured and extracted with 3 M HCl (2 mL), 2-propanol (1 mL), and MTBE/hexane (2 mL, 1:1, *v*:*v*). After KOH, the neutral phase was cleaned by using gel permeation chromatography and activated silica gel for column chromatography (Wakogel DX). The phenolic phase was acidified (pH 2) by using sulfuric acid and re-extracted by using MTBE/hexane (1:1, *v*:*v*). The phase was cleaned up by using 3 g of deactivated silica gel for column chromatography and methylated by using TMSDM. The derivatized solution was cleaned by using activated silica gel for column chromatography (Wakogel S-1).

Each cat food sample (3 g) was crushed into pieces, and the following internal standards were added: 3 M HCl (3 mL), 2-propanol (10 mL), and MTBE/hexane (20 mL, 1:1, *v*:*v*). This mixture was stirred with a homogenizer and extracted by ultrasonication. The cleanup procedures for the cat food were the same as those used in a previous study [5].

The chemical analysis of the house dust was slightly modified from that of a previous study [27]. Dust samples were air-dried in a 25 °C oven and were passed through a 250-µm mesh. The meshed dust (20–100 mg) was extracted by using an accelerated solvent extractor (SE100, Mitsubishi Chemical Analytech Co., Ltd., Tokyo, Japan) at a flow rate of 4 mL min^−1^ with MTBE/hexane (1:1, *v*:*v*) at 35 °C for 1 h. The extract was concentrated to 2 mL and was partitioned to neutral and phenolic phases. The cleanup methods for the organic phase were the same as those described previously [28]. The phenolic fraction was cleaned up by the same method as that used for the serum. This analytical method for dust samples was validated using certified dust material (NIST SRM2585) and the data obtained agreed well with the reference values [28].

The identification and quantification of PCBs, PBDEs, OH-PCBs, OH-PBDEs, and MeO-PBDEs were performed by using a gas chromatograph (GC: 6890 series, Agilent, Santa Clara, CA, USA) coupled with a high-resolution mass spectrometer (HRMS: JMS-800D, JEOL, Tokyo, Japan). The instrumental conditions are described in a previous report [27].

### 2.4. Analysis of Lipid Contents in Cat Sera

Most previous studies on cat sera reported OHC concentrations on a lipid weight basis. Therefore, the lipid contents of 26 randomly selected samples were analyzed according to the following formula for human sera [29] to facilitate a comparison between the OHC concentrations of Thai cat sera and those of other countries:TL = 1.33 × TG + 1.12 × CHOL + 1.48 (1)
where TL, TG, and CHOL represent the total lipids (%), triglycerides (µg dL^−1^), and cholesterol (µg dL^−1^), respectively, in the serum. Analyses of the triglycerides and cholesterol were entrusted to FUJIFILM VET Systems Co., Ltd. (Tokyo, Japan).

### 2.5. Quality Assurance/Quality Control

One procedural blank was analyzed in each batch (*n* = 7) to detect any possible contamination from solvents and glassware. Target compounds were identified and quantified according to the corresponding ^13^C_12_-internal standards by using the isotope dilution method reported in previous studies [27]. The method detection limit (MDL) was used as defined by the guidelines of the Ministry of the Environment, 2016 [30]. The blank tests (*n* = 7) were performed, the standard deviation calculated from the blank tests were compared to the standard deviation calculated from the environmental samples, and the larger value was used to calculate the MDL.

### 2.6. Estimation of Daily Intake

We calculated the daily intake (DI) of OHCs by pet cats by using the measured concentrations in exposure sources (wet weight for cat food and dry weight for dust) as follows:DI = (DC × C)/bw (2)
where DC, C, and bw are the daily ingestion of pet food or dust (g day^−1^), OHC concentration in the pet food or dust (ng g^−1^), and body weight of the pet cat (kg). The body weight (bw) was assumed as 4 kg, which was the average among the pet cats in this study. The daily ingestions of dry food and wet food were assumed as 120 and 425 g, respectively, which were the recommended amounts of the analyzed pet foods. Based on previous studies, the amount of dust ingested per day per cat was used a weight-adjusted value derived from U.S. EPA estimates of dust ingestion by toddlers (low: 50 mg or high: 200 mg) [31,32]. This is because there is currently no house dust exposure index for pet cats, so the amount of dust ingested by infants crawling and licking their fingers is used as a reference.

### 2.7. Statistical Analysis

For the statistical analysis, concentrations in the sample below the MDL were assumed to equal 0.5 × DL. Significant differences between the two groups were analyzed by using the Wilcoxon U test. A *p*-value of < 0.05 was considered to indicate statistical significance. Spearman’s rank correlation coefficients were calculated to assess the relationships between two components. All statistical analyses were conducted by using R program version 3.4.2 (http://www.r-project.org/ (accessed on 6 November 2022)).

## 3. Results

### 3.1. OHC Contamination in Cat Sera

Table 3 summarizes the OHC concentrations of all serum, cat food, and house dust samples. The details of the OHC concentrations are presented in Appendix A. The median concentrations (ranges in parentheses) of the PBDEs, OH-PBDEs, MeO-PBDEs, PCBs, and OH-PCBs in the Thai cat sera were 490 pg mL^−1^ (<MDL–48,000 pg mL^−1^), 77 pg mL^−1^ (9.9–440 pg mL^−1^), 19 pg mL^−1^ (<MDL–460 pg mL^−1^), 18 pg mL^−1^ (<MDL–1500 pg mL^−1^), and 59 pg mL^−1^ (1.9–1000 pg mL^−1^), respectively. There were no differences between the OHC concentrations for males and females, or between whether or not the females had birth experience. Therefore, the cat serum samples were not compared separately for males and females. Additionally, no significant differences were observed depending on the collection area (Bangkok, Nong Pho, and Hua Hin), so regional differences were not considered.

The cat sera had significantly higher levels of PBDEs than other contaminants. The median lipid-based concentration of PBDEs in the Thai cats (*n* = 26, median: 230 ng g^−1^ lipid in Appendix A) was comparable to that in cats from industrialized countries such as Japan and Sweden, but lower than in the USA [5,9,15,33]. These results indicate that pet cats in Thailand are chronically exposed to similar exposure sources as those of developed countries, excluding the USA. Meanwhile, the United States, reflecting past high-volume use, suggests continued ubiquitous pollution by PBDEs.

With regard to the PBDE profiles of the Thai cat sera, Figure 1 shows that BDE-209 was predominant followed by nona-BDEs (BDE-208, BDE-207, and BDE-206). The BDE-209 concentration (170 pmol g^−1^ lipid) was similar to those of cat sera from the USA (290 pmol g^−1^ lipid), Japan (77 pmol g^−1^ lipid), and Sweden (32 pmol g^−1^ lipid) [5,9,15,33]. The congener profiles of PBDEs were similar to those of Japanese cats [5] but different from those of the USA and Sweden, where BDE-47 and BDE-99 were predominant [9,15,33]. The differences in these PBDE profile can be attributed to differences in exposure sources. Dye et al. (2007) [9] detected PBDEs in dry food products, and BDE-209 was the dominant congener. Thus, high levels of BDE-209 in pet cat sera can be attributed to the consumption of dry cat food. In contrast, other studies showed that BDE-209 is a dominant congener in house dust in both Japan and the USA [34]. Thus, house dust may also be a source of high BDE-209 levels.

The PCB concentrations for the Thai cat sera (median: 10 ng g^−1^ lipid; Appendix A) were one order of magnitude lower than those reported for cats from developed countries such as Japan, the USA, Italy, and Spain, and they were similar to those in Turkey [5,33,35]. These results may be because the pet cats in our study were younger than the ones in the other studies. Previous studies have reported a positive correlation between age and PCB levels in the blood [15]. We also found a significant positive correlation between the age and the BDE-47 and PCB concentrations in the Thai cat sera (Appendix A). Studies on OHC contamination in the USA and Sweden targeted elderly pet cats (>12 years old) to verify the association with FHT. However, the serum samples in our study and the Turkish study were collected from younger cats (average of 4.1 years old). Alternatively, Thai pet cats may have less PCB exposure through the considered sources (i.e., pet foods and/or house dust).

The detection frequency of all PCB congeners was below 50%. CB-153 was detected in 46% of the serum samples, followed by CB-138, CB-28, and CB-180 (35%, 15%, and 15%, respectively). The PCB profiles were similar to those reported for other countries [5,15,23,36]. This suggests that there was no particular PCB contamination in recent years. Lower-chlorinated OH-PCBs such as 4′-OH-CB-25/26/4OH-CB-31, 4′-OH-CB-72, 4-OH-CB-70, and 4′-OH-CB-18 were predominant in the Thai cat sera, and the composition was consistent with the results of our previous study [5]. Cats preferentially metabolize lower-chlorinated PCBs and retain OH-PCBs in their blood due to the low activity of CYP2B-like enzymes [6].

The major OH-PBDE and MeO-PBDE congeners in the cat sera were 6-OH-BDE-47, 2′-OH-BDE-68, and 2′-MeO-BDE-68, which have natural origins. These OH-PBDEs have already been reported to be natural products in marine environments [5]. Pet cats in Thailand are exposed to not only artificial but also natural OHCs, which is similar to the results for cats in Japan, the UK, Sweden, and Pakistan [5,10,15,22]. We previously found high levels of 6-MeO-BDE-47 and 2′-MeO-BDE68 in cat food made from seafood. Domestic cats are exposed to MeO-PBDE through cat food containing fish, and OH-PBDEs in cat blood are produced by CYP-dependent demethylation of MeO-PBDEs [5]. These results indicate that Thai pet cats are also contaminated with OHCs similar to those in developed countries and need continued investigation into their exposure effects and health risks.

### 3.2. OHC Contamination in Cat Food Samples

Table 3 presents the concentrations of PCBs, PBDEs, MeO-PBDEs, and OH-PBDEs detected in the cat food from Thailand. The PBDE concentrations were higher in the dry foods than wet foods. In contrast, naturally occurring OHCs (MeO-PBDEs and OH-PBDEs) were predominant in the wet foods. The dry foods were mainly produced from meat, cereals, and rice; the wet foods were produced from fish (Table 2). These raw ingredients may have affected the contamination patterns. No target compounds were detected in a wet food made only from chicken breast (Wet 4 in Figure 2). The samples of wet foods mainly made from fish also showed differences in contamination patterns, which may be related to the type of fish used as the ingredient and the production area (including wild and farmed). PCBs and MeO-PBDEs were detected at higher concentrations in wet foods (Wet 1, Wet 3, and Wet 7) produced from tuna. PCBs and MeO-PBDEs were detected at high concentrations in higher trophic-level animals such as tuna, which indicates a dependence on the position of the ingredient within the food chain [37]. In contrast, except for Dry 7, there were no significant differences in contaminant concentrations between the samples of dry foods. This may be because dry foods from the same manufacturer were made from the same raw materials (abattoir offal from mainly chicken) even if they had different flavors (Table 2) [15].

BDE-209 was the predominant congener in the dry foods, yet BDE-47 and BDE-99 were predominant in the wet foods (Figure 1). Analysis of PBDEs in foodstuffs collected in France showed that BDE-47 and -99 accounted for 50–75% of the total PBDEs in fish, fish oil, and fishmeal, while BDE-209 accounted for approximately 90% of PBDEs in meat [38]. The PBDE profiles of the dry and wet cat foods were consistent with the PBDE profiles found in these food products. However, previous studies of dry dog food contamination have shown that BDE-209 contamination can occur during the manufacturing process [9]. This may explain why a relatively high concentration of BDE-209 was detected in the dry food samples. However, no studies have yet quantified flame retardant contamination of human or pet food during the manufacturing process.

The PBDE and PCB concentrations in the dry and wet foods from Thailand were comparable to those reported for Japan and Sweden [5,15]. However, the PBDE concentration in the dry and wet foods from Thailand was lower than that in the USA [9]. In this study, the highest PBDE concentration was detected in a dry food made in the USA (Dry 7 in Figure 2). Considering these results, the OHC contamination of cat food may depend on the country of origin. However, this conclusion is not definitive because the sample size was small.

### 3.3. OHC Contamination in House Dust Samples

The concentrations of PCBs, OH-PCBs, OH-PBDEs, and MeO-PBDEs in the house dust were below the MDLs. The median PBDE concentration was 240 ng g^−1^ (26–590 ng g^−1^), and BDE-209 was the predominant congener (Table 1 and Figure 1). The PBDE concentrations in house dust from Thailand were lower than the concentrations from the USA (4200 ng g^−1^), Australia (1200 ng g^−1^), and the UK (10,000 ng g^−1^). They were comparable to those of other Southeast Asian countries (42 ng g^−1^ in Vietnam, 218 ng g^−1^ in the Philippines, and 120 ng g^−1^ in Indonesia) [10,39,40]. The values in this study were similar to those of Thai house dust collected in 2014 (median of 46 ng g^−1^, range of 6.6–2200 ng g^−1^) [41]. The concentrations of PBDE congeners excluding BDE-209 in the present study were lower than those previously reported for BDE-47, -99, -100, and -153 in house dust of Thailand that were collected in 2007 [25]. These BDE congeners were recently added to the list of hazardous substances in Thailand [25]. Regulations on the use of PBDEs during this period may be responsible for the decreased concentrations of these congeners. However, because BDE-209 has not been continuously investigated in Thailand, further investigation on the transition of indoor pollution is necessary. BDE-209 was recommended for inclusion in the Annex A list at the 11th Meeting of the Persistent Organic Pollutants Review Committee in 2015. Moreover, BDE-209 has been added to the Thailand Hazardous Substances List which is annexed to the Notification of Ministry of Industry on List of Hazardous Substances (No. 5) B.E. 2562, dated 16th October B.E. 2562 (2019). Despite these regulations, BDE-209 will likely continue to enter the environment in the near future through the disposal and degradation of products.

### 3.4. Estimated Exposure of Pet Cats to OHCs via Cat Food and House Dust and Risk Assessment

Figure 1 showed the PBDE compositions in cat sera, pet foods, and house dust from Thailand. The composition in the cat sera was similar to that in the dry foods and dust, which suggests that they were major sources of PBDE exposure. However, the sera had a greater proportion of nona-BDEs than these sources. In an administrative study on male Sprague–Dawley rats with BDE-209, the production of hepta- and nona-BDEs by debromination was observed [42]. Therefore, the nona-BDEs in the cat sera may have been partially produced by the debromination of BDE-209. The PBDE concentrations were significantly higher in sera from indoor cats than from outdoor cats (*p* < 0.05) and indoor/outdoor cats (*p* < 0.01). This result suggests that indoor cats are exposed to PBDEs via house dust contaminated by furniture and electric appliances. Previously, a significant correlation was shown between PBDE levels in house dust from residential areas and in pet cat sera, which suggests that house dust is a significant exposure source for cats [15].

On the other hand, no clear relationship was observed between dietary habits (dry or wet food) and the PBDE levels and profiles in the cat sera (Appendix A). One reason may be that none of the pet cats in this study ate only wet cat food. In the wet cat foods, BDE-47 was the main congener, which can be explained by fish being the main raw ingredient. In the USA, cats fed with wet food had a higher BDE-47 concentration in their sera than cats fed with dry food [9]. On the other hand, the composition of such PBDEs has not been found in the sera of Thai pet cats.

Pet food is an important exposure source of OHCs for pet cats, but so is the intake of house dust by grooming. In the present study, we calculated the estimated contributions of pet food and dust to each OHC assuming that the cats had a standard body weight of 4 kg. The exposure to PCBs and MeO-PBDEs was presumed to be primarily due to dietary intake (about 100%). Assuming a house dust intake of 50 mg day^−1^, cat food is estimated to account for 29.5–77.8% of the exposure to PBDEs (Figure 3A–C). Assuming a worst-case scenario for dust intake of 200 mg day^−1^, dietary intake contributed 9.5–46.6% (Figure 3D–F). Some studies have concluded that house dust is the primary source of PBDE exposure for pet cats [9,31,43]. However, our results suggest that dry food is also a significant source.

## 4. Conclusions

This study suggests that pet food is an important source of exposure of OHCs to pet cats in Thailand. In addition, the analysis of dry and wet foods indicated that pet food contamination could be characterized by the country of origin and raw ingredients. These results may provide information to help owners choose healthier pet foods. Moreover, they can be used to guide decision-making by pet owners and policymakers to reduce OHC exposure and contamination. However, few studies at worldwide that focus on relationship between factors such as ingredient and produced countries, and OHCs concentration in pet food. Comprehensive monitoring of OHCs contaminations in pet food distributed not only in Asia but also around the world and evaluation the risks to pets are necessary in the future.

## Figures and Tables

**Figure 1 animals-12-03520-f001:**
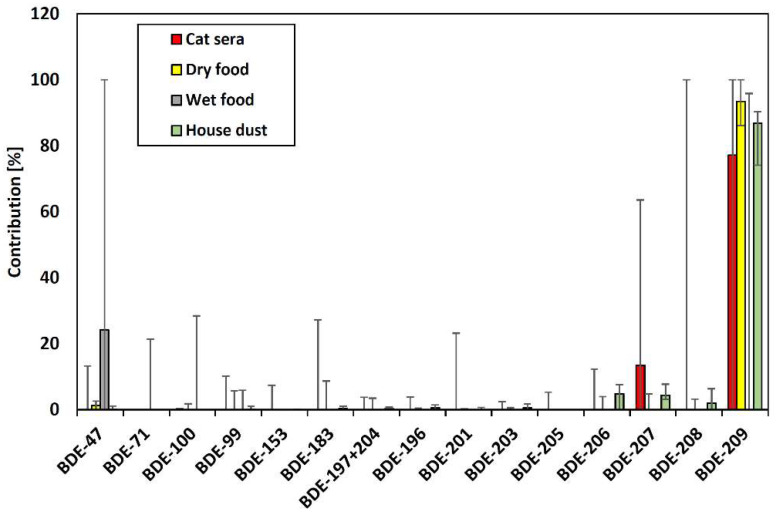
PBDE contributions in cat sera, cat food, and house dust in Thailand. Median and error bars indicate range of minimum to maximum.

**Figure 2 animals-12-03520-f002:**
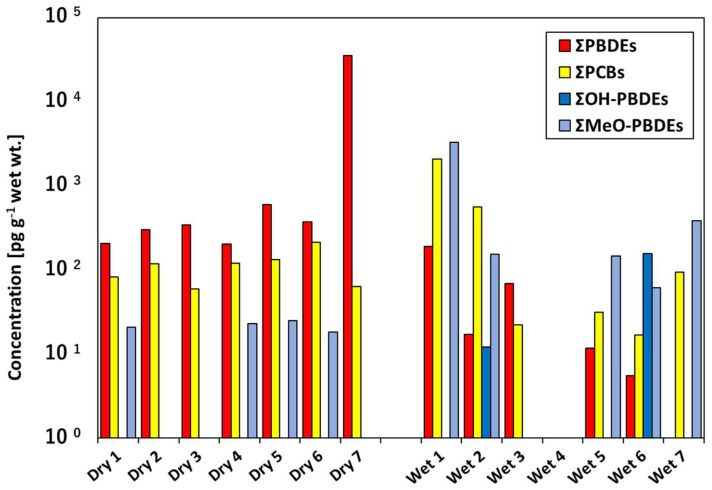
OHC concentrations in each dry and wet cat food.

**Figure 3 animals-12-03520-f003:**
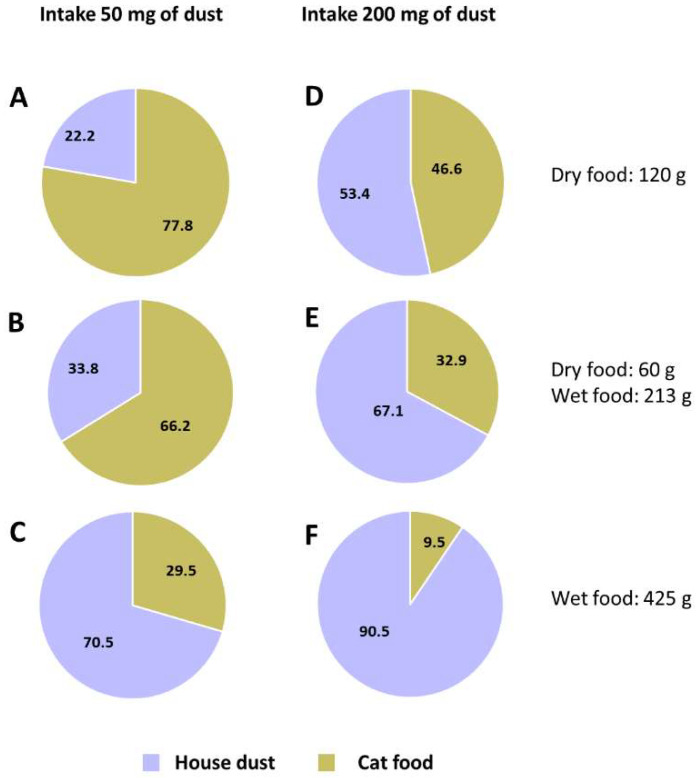
Estimating the contributions of cat food and house dust to PBDE exposure of pet cats in Thailand. These contributions were calculated by assuming a cat with a standard body weight of 4 kg. A pet cat was assumed to ingest dust (50 mg) and consume (**A**) dry food only, (**B**) half dry food and half wet food, or (**C**) wet food only. A pet cat was assumed to ingest dust (200 mg) and consume (**D**) dry food only, (**E**) half dry food and half wet food, and (**F**) wet food only.

**Table 1 animals-12-03520-t001:** Sample information of pet cats obtained from the questionnaires.

Sample ID	Hospital Location	Lipid Content [%] *^1^	Age [Year]	Weight [kg]	Breed	Sex	Diet	Housing	Q1 *^2^	Q2 *^3^	Q3 *^4^	Q4 *^5^	Q5 *^6^
BK04	Bangkok	0.66	2.0	4.85	Domestic short hair	Female	Dry	Outdoor	14	0	0	0	no
BK05	Bangkok	0.31	2.0	3.4	Domestic short hair	Female	Dry	Indoor	14	0	0	0	no
BK06	Bangkok	0.24	3.5	7	Domestic short hair	Male	Dry, Wet	Both	21	7	0	0	yes
BK08	Bangkok	0.48	0.3	4.4	Domestic short hair	Male	Dry, Wet	Indoor	14	0	14	0	yes
BK09	Bangkok	0.24	4.25	5.15	Domestic short hair	Male	Dry, Wet	Indoor	21	7	0	0	yes
BK10	Bangkok	0.23	1.0	4.14	Domestic short hair	Male	Dry	Indoor	21	0	0	0	no
BK11	Bangkok	0.30	5.1	5.0	Persian	Male	Dry	Indoor	14	0	0	0	yes
BK19	Bangkok	0.40	5.0	8.0	Domestic short hair	Male	Dry, Wet	Both	20	4	2	0	yes
BK24	Bangkok	0.51	2.0	3.7	Domestic short hair	Female	Dry, Home, Wet	Both	14	0	1	7	no
BK34	Bangkok	0.41	3.0	7.2	Domestic short hair	Male	Dry, Home, Wet	Both	14	0	1	7	no
BKK001	Bangkok	NA *^7^	1.0	5.0	Domestic short hair	Female	Dry	Indoor	7	0	0	0	yes
BKK013	Bangkok	NA	1.0	4.0	Domestic short hair	Male	Dry	Indoor	21	0	0	0	no
BKK017	Bangkok	NA	1.0	4.0	Domestic short hair	Male	Dry	Indoor	21	0	0	0	no
HH02	Hau Hin	0.19	7.0	5.8	British short hair	Male	Dry, Wet	Indoor	7	1	0	0	yes
HH03	Hau Hin	0.22	1.2	2.6	Domestic short hair	Male	Dry	Indoor	14	0	0	0	yes
HH04	Hau Hin	0.23	0.3	1.6	Domestic short hair	Male	Dry	Indoor	14	0	0	0	yes
HH05	Hau Hin	NA	5.0	3.3	Domestic short hair	Female	Dry, Wet	Indoor	7	0	7	0	yes
HH10	Hau Hin	0.41	1.3	3.7	Domestic short hair	Male	Dry	Indoor	14	0	0	0	yes
HH12	Hau Hin	0.22	3.0	4.5	Domestic short hair	Female	Dry, Wet	Outdoor	8	4	0	0	yes
HH13	Hau Hin	0.14	3.2	1.0	Domestic short hair	Female	Dry, Wet	Indoor	30	10	15	0	no
HH14	Hau Hin	NA	2.0	3.8	Domestic short hair	Female	Dry, Wet	Indoor	30	10	15	0	no
HH15	Hau Hin	NA	5.0	3.2	Domestic short hair	Female	Dry, Wet	Outdoor	14	0	14	0	no
HH18	Hau Hin	0.31	3.0	5.1	Domestic short hair	Male	Wet, Home, Dry	Both	7	0	14	14	yes
HH24	Hau Hin	0.12	5.0	4.1	Domestic short hair	Female	Dry	Outdoor	14	0	0	0	no
HH25	Hau Hin	0.23	3.0	4.0	Domestic short hair	Male	Wet, Home, Dry	Both	7	0	14	14	yes
HH28	Hau Hin	0.18	10	5.2	Mixed	Female	Dry, Wet	Indoor	7	0	7	0	Yes
HH32	Hau Hin	0.15	3.8	1.5	Maine coon	Female	Dry	Indoor	7	0	0	0	Yes
HH34	Hau Hin	0.29	3.0	2.2	Domestic short hair	Female	Dry, Home	Outdoor	2	0	0	2	No
HH35	Hau Hin	0.12	4.0	5.2	Unknown	Male	Dry, Wet, Home	Both	7	3	0	2	No
HH37	Hau Hin	NA	0.5	3.5	Unknown	Male	Dry	Indoor	7	0	0	0	No
HH41	Hau Hin	0.56	7.0	6.2	Mixed	Male	Wet, Dry, Home	Both	7	0	14	7	Yes
NP01	Nong Pho	NA	3.0	2.2	Persian	Female	Dry	Indoor	14	0	0	0	yes
NP02	Nong Pho	NA	7.0	4.4	Domestic short hair	Male	Dry, Wet	Both	14	2	2	0	yes
NP05	Nong Pho	NA	0.6	3.8	Domestic short hair	Male	Dry, Home, Wet	Indoor	14	1	0	14	yes
NP07	Nong Pho	NA	1.0	3.0	Domestic short hair	Female	Dry	Indoor	14	0	0	0	no
NP10	Nong Pho	NA	0.5	2.7	Domestic short hair	Female	Dry, Home	Both	21	0	0	7	no
NP11	Nong Pho	NA	0.5	2.5	Domestic short hair	Female	Home	Both	0	0	0	21	no
NP15	Nong Pho	NA	0.7	3.8	Domestic short hair	Male	Dry	Outdoor	35	0	0	0	yes
NP16	Nong Pho	NA	0.7	2.5	Domestic short hair	Female	Dry, Wet	Indoor	6	1	1	0	yes
NP18	Nong Pho	NA	0.7	2.5	Domestic short hair	Female	Dry, Home	Indoor	14	0	0	3	yes
NP20	Nong Pho	NA	0.9	4.0	Domestic short hair	Male	Dry, Home, Wet	Indoor	21	3	0	7	yes

*^1^ Lipid contents were measured in this study (see Section 2.4). *^2^ Q1: How many times do you give dry food in a week? *^3^ Q2: How many times do you give canned food in a week? *^4^ Q3: How many times do you give semi-moist food (pouch) in a week? *^5^ Q4: How many times do you give the same meal as the owner in a week? *^6^ Q5: Do you use cat litter? *^7^ NA: Not Analyzed.

**Table 2 animals-12-03520-t002:** Sample information of cat food written on the package labels purchased in Thailand.

Type	Sample ID	Produced Country	Maker	Main Ingredient
Dry	Dry 1	France	Maker A	Poultry meat, Rice, Animal fat, Corn, Fish oil
Dry	Dry 2	France	Maker A	Poultry meat, Rice, Wheat gluten, Animal fat, Corn, Fish oil
Dry	Dry 3	France	Maker A	Poultry meat, Wheat gluten, Rice, Corn, Animal fat, Flour
Dry	Dry 4	France	Maker A	Poultry meat, Wheat gluten, Rice, Corn, Animal fat, Flour
Dry	Dry 5	Thailand	Maker B	Cereal, Byproduct-meal, Vegetable protein, Fish meal, Tuna-byproduct
Dry	Dry 6	Thailand	Maker B	Cereal, Byproduct-meal, Vegetable protein, Fish meal, Shrimp-byproduct
Dry	Dry 7	USA	Maker C	Chicken meal, Corn, Rice, Fish oil
Wet (Canned)	Wet 1	Thailand	Maker D	Tuna, Shrimp
Wet (Canned)	Wet 2	Thailand	Maker B	Pilchards, Shrimp
Wet (Pouch)	Wet 3	Thailand	Maker B	Mackerel, Vegetable oil
Wet (Canned)	Wet 4	Thailand	Maker E	Chicken breast
Wet (Pouch)	Wet 5	Thailand	Maker E	Tuna, Ship jack tuna, Mackerel, Beef
Wet (Canned)	Wet 6	Thailand	Maker F	Tuna, Ocean fish, Vegetable oil,
Wet (Pouch)	Wet 7	Thailand	Maker F	Mackerel, Soybean oil

**Table 3 animals-12-03520-t003:** OHC concentration in cat serum, cat food, and house dust.

	Cat Serum (pg mL^−1^ Serum)		Dry Food (pg g^−1^ Wet wt.)		Wet Food (pg g^−1^ Wet wt.)		House Dust (pg g^−1^ Dry wt.)	
Compound	Median (Range)	DF (%)	Median	DF (%)	Median	DF (%)	Median	DF (%)
BDE-47	<MDL (<MDL–70)	26	4.9 (<MDL–15)	71	5.6 (<MDL–46)	71	260 (110–890)	100
BDE-99	<MDL (<MDL–95)	21	<MDL (<MDL–12)	29	<MDL (<MDL–11)	14	210 (130–1400)	100
BDE-209	350 (<MDL–37,000)	79	330 (180–31,000)	100	<MDL (<MDL–69)	29	200,000 (21,000–520,000)	100
ΣPBDEs	490 (<MDL–48,000)	100	350 (200–36,000)	100	12 (<MDL–190)	71	240,000 (26,000–590,000)	100
CB-153	8 (<MDL–340)	51	24 (20–43)	100	12 (<MDL–310)	71	<MDL (<MDL)	0
CB-138	<MDL (<MDL–250)	49	19 (15–32)	100	9.1 (<MDL–240)	71	<MDL (<MDL)	0
ΣPCBs	18 (<MDL–1500)	74	120 (59–210)	100	31 (<MDL–2100)	86	<MDL (<MDL–600)	29
4′-OH-CB-25/26/4-OH-CB-31	7 (<MDL–270)	82	<MDL (<MDL)	0	<MDL (<MDL)	0	<MDL (<MDL)	0
4′-OH-CB-72	14 (<MDL–53)	82	<MDL (<MDL)	0	<MDL (<MDL)	0	<MDL (<MDL)	0
ΣOH-PCBs	59 (1.9–1000)	100	<MDL (<MDL)	0	<MDL (<MDL)	0	<MDL (<MDL–95)	14
2′-OH-BDE-68	16 (<MDL–190)	74	<MDL (<MDL)	0	<MDL (<MDL–120)	14	<MDL (<MDL)	0
6-OH-BDE-47	55 (9.9–370)	100	<MDL (<MDL)	0	<MDL (<MDL–35)	29	<MDL (<MDL)	0
ΣOH-PBDEs	77 (9.9–440)	100	<MDL (<MDL)	0	<MDL (<MDL–160)	29	<MDL (<MDL)	0
2′-MeO-BDE-68	14 (<MDL–130)	59	<MDL (<MDL)	0	71 (<MDL–2800)	71	<MDL (<MDL)	0
6-MeO-BDE-47	<MDL (<MDL–380)	28	18 (<MDL–25)	57	17 (<MDL–420)	57	<MDL (<MDL)	0
ΣMeO-PBDEs	19 (<MDL–460)	62	18 (<MDL–25)	57	150 (<MDL–3300)	71	<MDL (<MDL)	0

DF: Detection frequency, MDL: Method detection limit.

## Data Availability

Not applicable.

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
