# Peer review of "Contamination Status of Pet Cats in Thailand with Organohalogen Compounds (OHCs) and Their Hydroxylated and Methoxylated Derivatives and Estimation of Sources of Exposure to These Contaminants"

_animals, 2022, doi:10.3390/ani12243520_

Round 1

Reviewer 1 Report

Major Comments.

I.               Being able to detect an organohalogen in sera, food, dust is not the same as being toxic. I wonder if the risk assessment is performed correctly.

1)    Author mentioned that none of the cats was diagnosed with hyperthyroidism or diabetes. Would measuring organohalogens in the serum of disease-free animals lead to a correct risk assessment? It would be better to compare data from healthy and diseased animals.

2)    Human data appear to be used in risk assessment. Author should consider species differences because expression of drug-metabolizing enzymes in cats may differ from that in other animal species. In addition, it is assumed that cats will orally ingest organohalogens such as dust and pet food. It seems to be necessary to consider the rate of absorption from the intestinal tract as a risk assessment.

II.           The explanation of sources of food and dust contamination is not clear.

  1)    Authors mentioned e-waste as a source of contamination, but is there any relationship between the geographical relationship of e-waste and the contamination of the seafood used as feed ingredients? It would be better to cite the paper.

  2)    It is not clear why the house dust was collected from the three regions, Bangkok, Sukhotai, Samut Songkhram and why the serum of cat was collected from the three region, Bangkok, Nong Pho and Hua Hin.

   Were the dust and cat serum collected from the same area?

III.             In addition, was the food that the serum-tested cat ate the same as the food sold at the local supermarket where the organohalogens were measured?

Round 2

Reviewer 1 Report

The authors addresses some of my concerns. However, it seems that issue I mentioned in my first review may not be appropriately addressed: 

1)     Authors mentioned e-waste as a source of contamination, but is there any relationship between the geographical relationship of e-waste and the contamination of the seafood used as feed ingredients? It would be better to cite the paper.

Ø  Line 93: We have added references: ~ the environment during the waste treatment [24].

Why did authors cite this paper?

I would appreciate it if you could explain a little more carefully.

Muenhor, D.; Harrad, S. Polybrominated diphenyl ethers (PBDEs) in car and house dust from Thailand: Implication for human exposure. J. Environ. Sci. Health Part A 2018, 53, 629-642.

Is this article suitable for answering question? 

Author Response

Response to Reviewer 1 Comments

The authors addresses some of my concerns. However, it seems that issue I mentioned in my first review may not be appropriately addressed: 

1)     Authors mentioned e-waste as a source of contamination, but is there any relationship between the geographical relationship of e-waste and the contamination of the seafood used as feed ingredients? It would be better to cite the paper.

Ø  Line 93: We have added references: ~ the environment during the waste treatment [24].

Why did authors cite this paper?

I would appreciate it if you could explain a little more carefully.

Muenhor, D.; Harrad, S. Polybrominated diphenyl ethers (PBDEs) in car and house dust from Thailand: Implication for human exposure. J. Environ. Sci. Health Part A 2018, 53, 629-642.

Is this article suitable for answering question? 

> Thank you for your comment. We are responding to the reviewer's point of view.

We have added the following sentence (Line 91~94) and a new reference [24].

In the reference paper, the spatial distribution of PBDEs suggests that inland sources may be affecting coastal areas.

  1. Kwan, C. S.; Takada, H.; Mizukawa, K.; Rinawati, M. S.; Santiago, E. C. Sedimentary PBDEs in urban areas of tropical Asian countries. Pollut. Bull. 2013, 76, 95-105.

Line 91~94: “PBDE concentrations in sediments in Southeast Asian countries were generally higher than those reported for industrialized countries, and the spatial distribution of PBDEs suggested that inland sources may impact coastal areas [24]”
